# OUT-OF-DISTRIBUTION DETECTION USING LAYER-WISE UNCERTAINTY IN DEEP NEURAL NETWORKS

## ABSTRACT

In this paper, we tackle the problem of detecting samples that are not drawn from the training distribution, i.e., out-of-distribution (OOD) samples, in classification. Many previous studies have attempted to solve this problem by regarding samples with low classification confidence as OOD examples using deep neural networks (DNNs). However, on difficult datasets or models with low classification ability, these methods incorrectly regard in-distribution samples close to the decision boundary as OOD samples. This problem arises because their approaches use only the features close to the output layer and disregard the uncertainty of the features. Therefore, we propose a method that extracts the uncertainties of features in each layer of DNNs using a reparameterization trick and combines them. In experiments, our method outperforms the existing methods by a large margin, achieving state-of-the-art detection performance on several datasets and classification models. For example, our method increases the AUROC score of prior work (83.8%) to 99.8% in DenseNet on the CIFAR-100 and Tiny-ImageNet datasets.

## 1 INTRODUCTION

Deep neural networks (DNNs) have achieved high performance in many classification tasks such as image classification (Krizhevsky et al., 2012; Simonyan & Zisserman, 2014), object detection (Lin et al., 2017; Redmon & Farhadi, 2018), and speech recognition (Hinton et al., 2012; Hannun et al., 2014). However, DNNs tend to make high confidence predictions even for samples that are not drawn from the training distribution, i.e., out-of-distribution (OOD) samples (Hendrycks & Gimpel, 2016). Such errors can be harmful to medical diagnosis and automated driving. Because it is not generally possible to control the test data distribution in real-world applications, OOD samples are inevitably included in this distribution. Therefore, detecting OOD samples is important for ensuring the safety of an artificial intelligence system (Amodei et al., 2016).

There have been many previous studies (Hendrycks & Gimpel, 2016; Liang et al., 2017; Lee et al., 2017; DeVries & Taylor, 2018; Lee et al., 2018; Hendrycks et al., 2018) that have attempted to solve this problem by regarding samples that are difficult to classify or samples with low classification confidence as OOD examples using DNNs. Their approaches work well and they are computationally efficient. The limitation of these studies is that, when using difficult datasets or models with low classification ability, the confidence of inputs will be low, even if the inputs are in-distribution samples. Therefore, these methods incorrectly regard such in-distribution samples as OOD samples, which results in their poor detection performance (Malinin & Gales, 2018), as shown in Figure 1.

One cause of the abovementioned problem is that their approaches use only the features close to the output layer and the features are strongly related to the classification accuracy. Therefore, we use not only the features close to the output layer but also the features close to the input layer. We hypothesize that the uncertainties of the features close to the input layer are the uncertainties of the feature extraction and are effective for detecting OOD samples. For example, when using convolutional neural networks (CNNs), the filters of the convolutional layer close to the input layer extract features such as edges that are useful for in-distribution classification. In other words, in-distribution samples possess more features that convolutional filters react to than OOD samples. Therefore, the uncertainties of the features will be larger when the inputs are in-distribution samples.

Another cause of the abovementioned problem is that their approaches disregard the uncertainty of the features close to the output layer. We hypothesize that the uncertainties of the latent features close

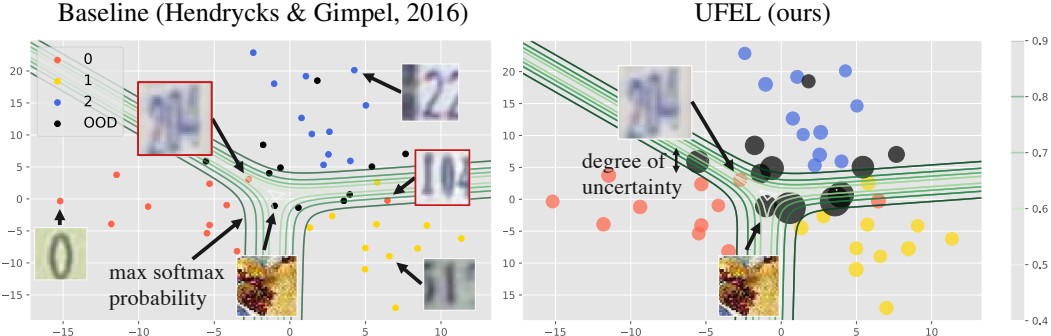

Figure 1: Comparison of existing and proposed methods. We visualized scatter plots of the outputs of the penultimate layer of a CNN that can estimate the uncertainties of latent features using the SVHN dataset (Netzer et al., 2011). We used only classes 0, 1, and 2 for the training data. Classes 0, 1, 2, and OOD, indicated by red, yellow, blue, and black, respectively, were used for the validation data. We plot the contour of the maximum output of the softmax layer of the model. **Left:** Because the image of "204" includes the digits "2" and "0," the maximum value of the softmax output decreases because the model does not know to which class the image belongs. **Right:** The sizes of points in the scatter plots indicate the value of the combined uncertainties of features. We can classify the image of "204" as an in-distribution image according to the value of the combined uncertainties.

to the output layer are the uncertainties of classification and are also effective for detecting OOD samples. For example, in-distribution samples are embedded in the feature space close to the output layer to classify samples. In contrast, OOD samples have no fixed regions for embedding. Therefore, the uncertainties of the features of OOD samples will be larger than those of in-distribution samples.

Based on the hypotheses, we propose a method that extracts the Uncertainties of Features in Each Layer (UFEL) and combines them for detecting OOD samples. Each uncertainty is easily estimated after training the discriminative model by computing the mean and the variance of their features using a reparameterization trick such as the variational autoencoder (Kingma & Welling, 2013) and variational information bottleneck (Alemi et al., 2016; 2018). Our proposal is agnostic to the model architecture and can be easily combined with any regular architecture with minimum modifications.

We visualize the maximum values of output probability and the combined uncertainties of the latent features in the feature space of the penultimate layer in Figure 1. The combined uncertainties of the features discriminate the in-distribution and OOD images that are difficult to classify. For example, although the images that are surrounded by the red line are in-distribution samples, they have low maximum softmax probabilities and could be regarded as OOD samples in prior work. Meanwhile, their uncertainties are smaller than those of OOD samples and they are regarded as in-distribution samples in our method.

In experiments, we validate the hypothesis demonstrating that each uncertainty is effective for detecting OOD examples. We also demonstrate that UFEL can obtain state-of-the-art performance in several datasets including CIFAR-100, which is difficult to classify, and models including LeNet5 with low classification ability. Moreover, UFEL is robust to hyperparameters such as the number of in-distribution classes and the validation dataset.

## 2  RELATED WORK

**Methods based on the classification confidence**  Hendrycks & Gimpel (2016) proposed the baseline method to detect OOD samples without the need to further re-train and change the structure of the model. They define low-maximum softmax probabilities as indicating the low confidence of in-distribution examples and detect OOD samples using the softmax outputs of a pre-trained deep classifier. Building on this work, many models have recently been proposed. Liang et al. (2017) proposed ODIN, a calibration technique that uses temperature scaling (Guo et al., 2017) in the

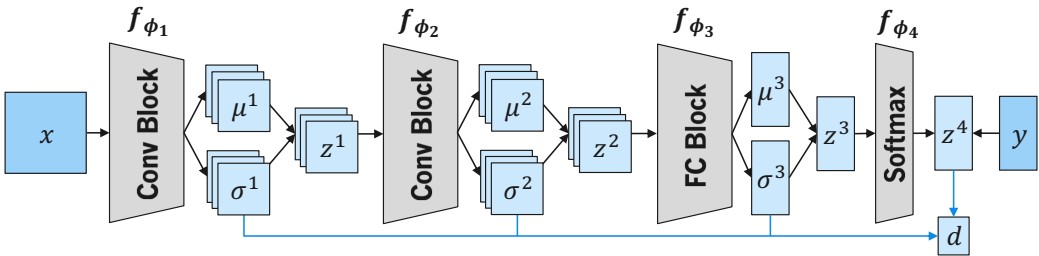

Figure 2: Network structure of UFEL when using DenseNet. **Black arrow:** Extracting the variance of latent features using the reparameterization trick. **Blue arrow:** Combining these features.

softmax function and adds small controlled perturbations to the inputs to widen the gap between in-distribution and OOD features, which improves the performance of the baseline method. Likewise, Lee et al. (2018); DeVries & Taylor (2018); Lee et al. (2017); Hendrycks et al. (2018) also extended the baseline method. Like Hendrycks & Gimpel (2016), we use the feature of maximum softmax probability as one of our features.

**Methods based on the uncertainty** Malinin & Gales (2018) attempted to solve the problem of classifying in-distribution samples close to the decision boundary as OOD samples by distinguishing between *data uncertainty* and *distributional uncertainty*. Data uncertainty, or aleatoric uncertainty (Kendall & Gal, 2017), is irreducible uncertainty such as class overlap, whereas distributional uncertainty arises because of the mismatch between training and testing distributions. They argue that the value of distributional uncertainty depends on the difference in the Dirichlet distribution of the categorical parameter. Further, they estimate the parameter of the Dirichlet distribution using a DNN and train the model with in-distribution and OOD datasets. The motivation for our work is similar to that of Malinin & Gales (2018). In our work, the distribution of the logit of the categorical parameters is modeled as a Gaussian distribution, which enables us to train the model without an OOD dataset. Furthermore, we estimate the parameters of the Gaussian distribution of latent features close to the input layer.

## 3 PROPOSED METHOD

In this section, we present UFEL, which extracts the uncertainties of features in each layer and combines them for detecting OOD samples. First, we use the maximum of the softmax output, as in Hendrycks & Gimpel (2016), as one of our features. Second, we also use the distribution of the categorical parameter, as in Malinin & Gales (2018), using the uncertainty of logits. Furthermore, we use the uncertainty of the feature extraction extracted from the latent space close to the input layer because they will not be relevant to the classification accuracy. We probabilistically model the values of these features, estimate their uncertainties, and combine them.

Let $x \in \mathcal{X}$ be an input, $y \in \mathcal{Y} = \{1, \cdots, K\}$ be its label, and $l \in \{1, \cdots, L\}$ be the index of the block layers. The objective function of normal deep classification is as follows:

$$\mathcal{J} = \mathbb{E}_{x,y \sim p(x,y)}[\mathcal{L}(f_\phi(x), y)], \tag{1}$$

where $p(x, y)$ is the empirical data distribution, $\mathcal{L}$ is a cross entropy loss function, and $f_\phi$ is a DNN. We use the following notation $f_\phi = f_{\phi_L} \circ f_{\phi_{L-1}} \circ \cdots \circ f_{\phi_1}$ as shown in Figure 2.

To extract the uncertainties of features in each layer, we model the $l$th block layer's output $z^l$ as a Gaussian whose parameters depend on the $l$-1th block layer's output $z^{l-1}$ as follows: $p(z^l|z^{l-1}) = \mathcal{N}(z^l|f_{\phi_l}^\mu(z^{l-1}), f_{\phi_l}^\Sigma(z^{l-1}))$, where $f_{\phi_l}$ is the $l$th block layer, which outputs both the mean $\mu$ and covariance matrix $\Sigma$. In this paper, we use a diagonal covariance matrix to reduce the model parameters. We use the reparameterization trick (Kingma & Welling, 2013) to write $z^l = \mu^l + \sigma^l \epsilon$, where $\mu^l = f_{\phi_l}^\mu(z^{l-1})$, $\sigma_l = f_{\phi_l}^\Sigma(z^{l-1})$, and $\epsilon$ is the Gaussian noise. Then, our objective function is as follows:

$$\mathcal{J}_\mathcal{U} = \mathbb{E}_{x,y \sim p(x,y), \epsilon \sim p(\epsilon)}[\mathcal{L}(z^L(x, \epsilon), y)], \tag{2}$$

where $z^0 = x$. Because of the reparameterization trick, the loss gradient is backpropagated directly through our model, and we can train our model like the regular classification models in Equation 1.

Next, we explain the two methods of combining the features extracted in each layer. In the first method, we sum the uncertainties of each value of the features in each layer and linearly combine them. Because the feature maps of a convolutional block layer are three dimensional, each element is computed as $z_{ijk}^l = \mu_{ijk}^l + \sigma_{ijk}^l \epsilon$. Moreover, because the output of a fully connected layer is one dimensional, each element is formed as $z_i^l = \mu_i^l + \sigma_i^l \epsilon$. We use a weighted summation of the scale of each feature and the maximum value of the softmax scores as a final feature $d_{LR}$ as follows:

$$d_{LR}(x) = \sum_{l=1}^{L-1} \lambda^l \sigma_{sum}^l(x) + \lambda^L \max(z^L), \sigma_{sum}^l = \begin{cases} \sum_{ijk} \sigma_{ijk}^l & \text{(if the } l\text{th block is Conv)} \\ \sum_i \sigma_i^l & \text{(otherwise)} \end{cases} \quad (3)$$

We choose the parameter $\lambda^l$ by training a logistic regression (LR) using in-distribution and OOD validation samples. In the second method, we combine the features directly and nonlinearly using a CNN as follows:

$$d_{CNN}(x) = CNN_\theta(\sigma^1(x), ..., \sigma^{L-1}(x), \max(z^L)). \quad (4)$$

We train the CNN parameter $\theta$ with in-distribution and OOD validation samples using binary crossentropy loss. The detailed structures of the CNN are given in Table A.3. We use the values of these feature $d(x)$ to test the performance of detecting OOD samples.

## 4 EXPERIMENTAL SETUP

In this section, we present the details of the experiments, which includes the datasets, metrics, comparison methods, and models. Because of space limitations, more details are given in Appendix A.

**Datasets** We used several standard datasets for detecting OOD samples and classifying in-distribution samples. The SVHN, CIFAR-10, and CIFAR-100 datasets were used as in-distribution datasets, whereas Tiny ImageNet (TIM), LSUN, iSUN, Gaussian noise, and uniform noise were used as OOD datasets. These data were also used in Liang et al. (2017); DeVries & Taylor (2018). We applied standard augmentation (cropping and flipping) in all experiments. We used 5,000 validation images split from each training dataset and chose the parameter that can obtain the best accuracy in the validation dataset. We also used 68,257 training images from the SVHN dataset and 45,000 training images from the CIFAR-10 and CIFAR-100 datasets. All the hyperparameters of ODIN and UFEL were tuned on a separate validation set, which consists of 100 OOD images from the test dataset and 1,000 images from the in-distribution validation set. We tuned the parameters of the CNN in Equation 4 using 50 validation training images taken from the 100 validation images. The best parameters were chosen by validating the performance using the rest of 50 validation images. Finally, we tested the models with a test dataset that consisted of 10,000 in-distribution images and 9,900 OOD images.

**Evaluation metrics** We used several standard metrics for testing the detection of OOD samples and the classification of in-distribution samples. We used TNR at 95% TPR, AUROC, AUPR, and accuracy (ACC), which were also used in Lee et al. (2017; 2018).

**Comparison method** We compare UFEL with the baseline (Hendrycks & Gimpel, 2016) and ODIN (Liang et al., 2017) methods. For the baseline method, we used $\max_k p(y = k|x)$ as the detection metric. For ODIN, we used the same detection metric and calibrated it using temperature scaling and small perturbations to the input. The temperature parameter $T \in \{1, 10, 100, 1000\}$ and the perturbation parameter $\epsilon \in \{0, 0.001, 0.005, 0.01, 0.05, 0.1\}$ were chosen using the in-distribution and OOD validation datasets.

**Model training details** We adopted LeNet5 (LeCun et al., 1998) and two state-of-the-art models, WideResNet (He et al., 2016) and DenseNet (Huang et al., 2017), in this experiment. In all experiments, we used the same model and conditions to compare UFEL with existing methods. Only the structure used to extract the variance parameters differs. For LeNet5, we increased the number of channels of the original LeNet5 to improve accuracy. See Table A.3 for model details. We inserted

Table 1: Results for the OOD detection test set data for image classification when the in-distribution dataset is CIFAR-100 and the OOD dataset is iSUN. We trained Dense-BC for each method under the same conditions. All results are averaged over three runs ± one standard deviation. All values are percentages, and the best results are indicated in bold.

| Method | ACC | TNR at TPR 95% | AUROC | AUPR-In | AUPR-Out |
|--------|-----|----------------|-------|---------|----------|
| Baseline | 71.1±0.2 | 10.5±1.6 | 61.7±5.4 | 65.4±6.5 | 57.1±3.9 |
| ODIN | | 43.4±10.4 | 85.1±5.7 | 86.6±5.8 | 82.9±6.0 |
| max P | 71.6±0.4 | 12.3±1.7 | 63.9±4.1 | 67.8±4.9 | 59.2±2.9 |
| var 1 | | 79.7±3.8 | 93.9±0.9 | 92.8±0.8 | 94.7±0.9 |
| var 2 | | 35.2±10.0 | 79.4±2.9 | 79.8±2.0 | 77.5±4.6 |
| var 3 | | 54.7±10.9 | 90.8±2.5 | 92.4±2.1 | 87.4±3.4 |
| UFEL (LR) | | 93.1±0.7 | 98.6±0.1 | 98.7±0.1 | 98.5±0.1 |
| UFEL (CNN) | | **99.3**±0.5 | **99.8**±0.1 | **99.8**±0.1 | **99.8**±0.1 |

the reparameterization trick to the second convolutional layer and the softmax layer. LeNet5 was trained using the Adam (Kingma & Ba, 2014) optimizer for 10 epochs and the learning rate was set to 5e-4. Both DenseNet and WideResNet were trained using stochastic gradient descent, with a Nesterov momentum of 0.9. We inserted the reparameterization trick to the first convolutional block, the second convolutional block, and the softmax layer. For WideResNet, we used a WideResNet with a depth of 40 and width of 4 (WRN-40-4), which was trained for 50 epochs. The learning rate was initialized to 0.1 and reduced by a factor of $10\times$ after the 40th epoch. For DenseNet, we used a DenseNet with depth $L = 100$ (Dense-BC), growth rate of 12, and drop rate of 0. DenseNet-BC was trained for 200 epochs with batches of 64 images, and a weight decay of 1e-4 for the CIFAR-10 and CIFAR-100 datasets. It was trained for 10 epochs for the SVHN dataset. The learning rate was initialized to 0.1 and reduced by a factor of $10\times$ after the 150th epoch.

## 5 EXPERIMENTAL RESULTS

In this section, we demonstrate the performance of UFEL by conducting five experiments. In the first experiment, we show that UFEL performs better than the baseline (Hendrycks & Gimpel, 2016) and ODIN (Liang et al., 2017) methods on several datasets and models. In the second experiment, we confirm that the features of UFEL have almost no relationship with the ACC. In the third experiment, we demonstrate that UFEL has a strong ability to detect OOD data, even if the number of classes of in-distribution data is small. In the fourth experiment, we confirm that UFEL is robust to the number of OOD samples, and in the fifth experiment, we test the performance of UFEL on unseen OOD datasets. The objective of these experiments is to show the uncertainties of the features obtained in each CNN layer distinguish the in-distribution and OOD data. Moreover, we obtain state-of-the-art performance for OOD sample detection by combining these features.

**Detecting OOD samples on several datasets and models** In this experiment, we evaluate the performance of OOD detection using Equation 3 and Equation 4. In this study, var $l$ is used to denote $\sigma_{sum}^{l}$, max P is $\max(z^{L})$, UFEL (LR) denotes $d_{LR}$ in Equation 3, and UFEL (CNN) denotes $d_{CNN}$ in Equation 4. We measured the detection performance using a DenseNet trained on CIFAR-100 when the iSUN dataset is used to provide the OOD images. Table 1 shows that var 1 and var 3 are strong features that, by themselves, can outperform ODIN. This indicates that the uncertainties of the feature extraction and classification are effective for detecting OOD samples. Moreover, the combination of these features yields state-of-the-art performance.

In Table 2, we demonstrate that UFEL outperforms the baseline and ODIN methods on several datasets and models. Furthermore, UFEL is also slightly superior to them with respect to in-distribution accuracy, which indicates that our model is robust to noise because of the reparameterization trick. Here, we do not report ODIN accuracy because the model of ODIN is the same as that of the baseline. We conducted this experiment three times and used the average of the results. We used the CIFAR-10, CIFAR-100, and SVHN datasets as the in-distribution datasets and the other datasets as the OOD samples. Note that our UFEL outperformed the baseline and ODIN methods by a large margin, especially when using CIFAR-100, which is difficult to classify, or LeNet5 which

Table 2: Results for the OOD detection test set data for various situations. All results are averaged over three runs. All values are percentages, and the best results are indicated in bold.

| In-dist (model) Baseline / UFEL (ACC) | OOD | TNR at TPR 95% | AUROC |
|---|---|---|---|
| | | Baseline / ODIN / UFEL (LR) / UFEL (CNN) | |
| SVHN (LeNet5) 90.7 / 91.2 | TIM | 43.8 / 68.1 / 95.7 / **99.1** | 89.7 / 93.0 / 98.8 / **99.8** |
| | LSUN | 42.3 / 66.0 / 98.4 / **99.8** | 89.2 / 92.7 / 99.6 / **99.9** |
| | iSUN | 46.8 / 72.0 / 97.7 / **99.3** | 90.6 / 94.1 / 99.4 / **99.8** |
| | Gaussian | 99.9 / **100.0/100.0/100.0** | 99.9 / **100.0/100.0/100.0** |
| | Uniform | 98.7 / **100.0/100.0/100.0** | 99.6 / 99.9 / **100.0/100.0** |
| SVHN (Dense-BC) 95.8 / 95.9 | TIM | 72.4 / 84.3 / 99.4 / **99.7** | 95.8 / 97.1 / 99.8 / **99.9** |
| | LSUN | 67.6 / 80.5 / **99.9 / 99.9** | 95.0 / 96.4 / 99.9 / **100.0** |
| | iSUN | 73.6 / 85.5 / 99.8 / **99.9** | 95.9 / 97.3 / 99.9 / **100.0** |
| | Gaussian | 81.2 / 92.4 / 99.8 / **100.0** | 97.2 / 98.3 / 99.7 / **100.0** |
| | Uniform | 90.2 / 97.7 / **100.0/100.0** | 98.2 / 99.1 / 99.9 / **100.0** |
| CIFAR-10 (LeNet5) 72.6 / 73.1 | TIM | 7.5 / 8.6 / 48.7 / **92.3** | 58.8 / 59.7 / 83.7 / **97.7** |
| | LSUN | 5.7 / 6.4 / 62.6 / **98.0** | 56.9 / 58.4 / 90.9 / **99.5** |
| | iSUN | 6.1 / 7.2 / 53.9 / **95.7** | 57.2 / 59.0 / 86.0 / **98.9** |
| | Gaussian | 52.8 / 80.3 / 48.3 / **100.0** | 93.3 / 96.6 / 89.9 / **100.0** |
| | Uniform | 48.7 / 55.8 / 88.5 / **100.0** | 91.9 / 94.4 / 97.5 / **100.0** |
| CIFAR-10 (WRN-40-4) 88.8 / 89.6 | TIM | 25.8 / 72.7 / 93.2 / **99.4** | 84.6 / 95.2 / 98.5 / **99.8** |
| | LSUN | 31.0 / 84.2 / 96.9 / **100.0** | 87.5 / 97.1 / 99.3 / **100.0** |
| | iSUN | 30.3 / 82.9 / 94.1 / **99.9** | 87.1 / 97.0 / 98.6 / **100.0** |
| | Gaussian | 36.2 / 90.1 / **100.0/100.0** | 90.4 / 98.2 / 99.9 / **100.0** |
| | Uniform | 58.0 / 96.2 / **100.0/100.0** | 94.7 / 99.0 / 99.8 / **100.0** |
| CIFAR-10 (Dense-BC) 92.1/92.4 | TIM | 40.0 / 85.0 / 90.6 / **96.4** | 90.5 / 97.4 / 98.1 / **99.2** |
| | LSUN | 47.1 / 92.4 / 93.9 / **99.4** | 92.2 / 98.5 / 98.8 / **99.8** |
| | iSUN | 46.4 / 91.2 / 93.6 / **99.7** | 92.1 / 98.4 / 98.8 / **99.9** |
| | Gaussian | 37.2 / 97.3 / **100.0/100.0** | 92.6 / 99.2 / **100.0/100.0** |
| | Uniform | 53.8 / 92.7 / **100.0/100.0** | 94.4 / 98.7 / **100.0/100.0** |
| CIFAR-100 (LeNet5) 38.8 / 39.6 | TIM | 1.6 / 1.6 / 51.4 / **88.7** | 45.9 / 45.7 / 85.8 / **96.4** |
| | LSUN | 0.6 / 0.6 / 64.6 / **97.3** | 42.0 / 42.0 / 92.0 / **99.3** |
| | iSUN | 0.9 / 0.9 / 56.9 / **96.2** | 43.3 / 43.3 / 88.1 / **98.9** |
| | Gaussian | 0.0 / 0.0 / 37.2 / **100.0** | 42.6 / 47.9 / 93.1 / **100.0** |
| | Uniform | 0.2 / 0.0 / 99.2 / **100.0** | 50.2 / 56.5 / 97.6 / **100.0** |
| CIFAR-100 (WRN-40-4) 63.9 / 66.0 | TIM | 9.0 / 32.9 / 88.6 / **99.1** | 63.2 / 81.5 / 97.6 / **99.8** |
| | LSUN | 9.4 / 38.1 / 94.2 / **99.9** | 65.2 / 85.2 / 98.7 / **100.0** |
| | iSUN | 8.4 / 32.9 / 89.9 / **99.6** | 63.8 / 83.0 / 97.8 / **99.9** |
| | Gaussian | 6.8 / 39.2 / **100.0/100.0** | 60.2 / 89.0 / 99.6 / **100.0** |
| | Uniform | 4.8 / 34.1 / 92.6 / **100.0** | 63.5 / 87.0 / 97.7 / **100.0** |
| CIFAR-100 (Dense-BC) 71.1 / 71.6 | TIM | 11.5 / 43.3 / 87.9 / **98.9** | 61.5 / 83.8 / 97.3 / **99.8** |
| | LSUN | 11.3 / 46.0 / 95.5 / **99.9** | 62.4 / 86.2 / 99.0 / **100.0** |
| | iSUN | 10.5 / 43.4 / 93.1 / **99.3** | 61.7 / 85.1 / 98.6 / **99.8** |
| | Gaussian | 23.6 / 99.0 / **100.0/100.0** | 84.7 / 99.4 / **100.0/100.0** |
| | Uniform | 32.7 / 98.0 / **100.0/100.0** | 89.5 / 99.3 / **100.0/100.0** |

has lower classification ability. This result indicates that UFEL is less affected by in-distribution accuracy, and can discriminate between actual OOD samples and class-overlapping samples from in-distribution samples.

**Relationship between the performance of detecting OOD samples and in-distribution accuracy**
In this experiment, we show that the features of our method are not related to the in-distribution accuracy. We used CIFAR-10 as the in-distribution dataset and TIM as the OOD dataset. We trained DenseNet-BC for nine epochs and tested the performance at each epoch. As shown in Figure 3, each variance (var $l$) is less related to the accuracy than the baseline and ODIN methods. The var 1 of the feature close to the input layer has the highest ability to detect OOD samples in this experiment.

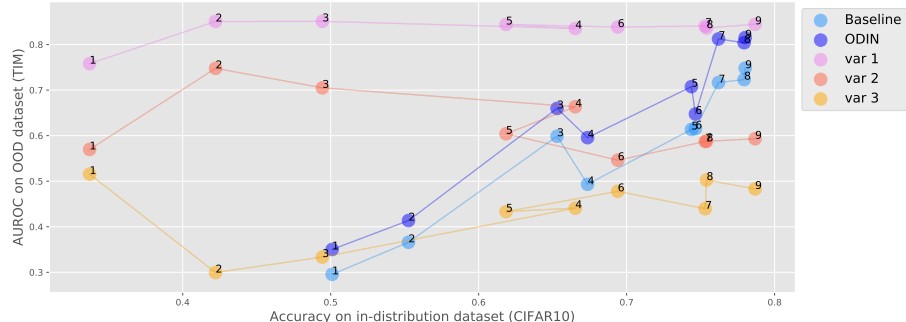

Figure 3: Plot of ACC (x-axis) and AUROC (y-axis). The number on the plot indicates the number of training epochs. We used CIFAR-10 (as in-distribution), TIM (as OOD), and the DenseNet-BC model. This graph shows that the AUROC of UFEL is less related to ACC than those of the baseline and ODIN.

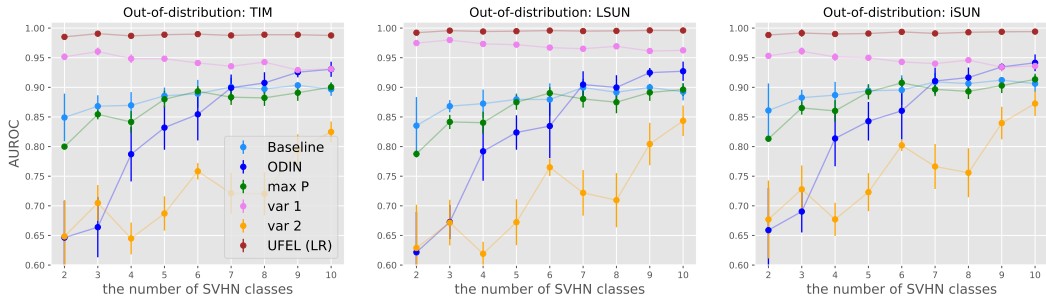

Figure 4: Plot of AUROC (y-axis) when changing the number of in-distribution dataset classes (x-axis). We used SVHN as in-distribution dataset, TIM, LSUN, and iSUN as OOD datasets, and the LeNet5 model. All plots were averaged over three runs and the error bar indicates one standard deviation.

These results also indicate that we can discriminate in-distribution and OOD examples when using a dataset that is difficult to classify.

**Detecting OOD samples while changing the number of in-distribution classes** In this experiment, we show that UFEL is robust to the number of class labels. We used SVHN as in-distribution dataset and changed the number of in-distribution classes in training as $\{0,1\}$, $\{0,1,2\}$,..., $\{0,1,2,...,9\}$. We also used TIM, LSUN and iSUN datasets as OOD samples, and LeNet5 as a model. We compared the proposed method with the baseline and ODIN methods, as shown in Figure 4. This graph shows the AUROC score of each model when changing the number of training data classes. As this graph shows, UFEL outperforms other methods in all cases and is robust to the number of in-distribution classes, whereas the performance of ODIN drops as the number of class labels decreases. These results suggest that UFEL is effective for small datasets because the number of samples can be decreased to one fifth of the original number when there are two in-distribution classes and the cost of label annotation is reduced.

**Detecting OOD samples while changing the number of OOD samples** In this experiment, we present the performance of UFEL while changing the number of OOD validation examples. All the hyperparameters of ODIN and UFEL were tuned on a separate validation set, which consists of 30, 50, and 100 OOD images in the test dataset and 1,000 images from the in-distribution validation set. As shown in Figure 5, although UFEL (CNN) outperforms other methods including UFEL (LR) in most cases, it performs worse than ODIN in part of the results because some tuning for OOD samples is needed. Meanwhile, UFEL (LR) outperforms prior methods constantly because the number of hyperparameters is small and tuning samples are almost unneeded.

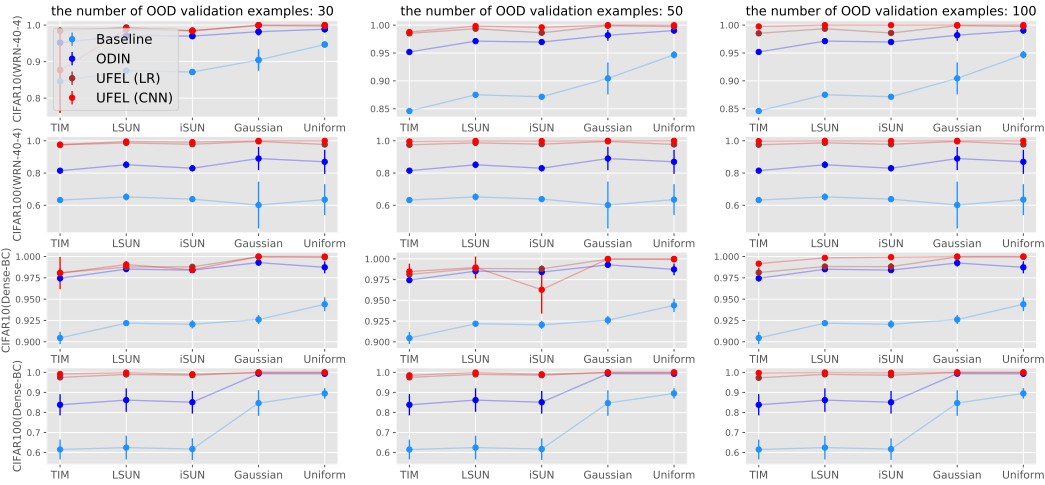

Figure 5: Plot of AUROC (y-axis) when changing the OOD dataset (x-axis). We used CIFAR-10 and CIFAR-100 as the in-distribution dataset. All plots are averaged over three runs and the error bar indicates one standard deviation.

**Generalization to unseen OOD dataset** Because OOD validation samples might not be available in practice, we used only uniform noise as the validation OOD dataset and tested the ability of our model to detect another OOD dataset. We added a binary classification as a comparison method. This method was trained using an in-distribution dataset (positive) and uniform noise (negative). Table 3 shows that UFEL outperforms prior work in all cases and generalize well. Table 3 also indicates that the binary classification method does not generalize well because it cannot distinguish in-distribution dataset and OOD datasets TIM, LSUN, and iSUN, although it can distinguish Gaussian noise, which is similar to uniform noise.

Table 3: AUROC score for OOD detection test set data. All results are averaged over three runs. All values are percentages and the best results are indicated in bold.

| AUROC | | WRN-40-4 | Dense-BC |
|---|---|---|---|
| In-dist | OOD | Binary / Baseline / ODIN / UFEL (LR) / UFEL (CNN) | |
| CIFAR-10 | TIM | 41.3 / 84.6 / 94.6 / 98.0 / **98.7** | 71.2 / 90.5 / 96.3 / 96.6 / **97.8** |
| | LSUN | 40.8 / 87.5 / 96.7 / 98.5 / **99.3** | 75.1 / 92.2 / 97.5 / 97.1 / **97.9** |
| | iSUN | 42.1 / 87.1 / 96.6 / 98.0 / **99.3** | 75.4 / 92.1 / 97.4 / 97.8 / **98.4** |
| | Gaussian | **100.0**/ 90.4 / 98.2 / 99.9 /**100.0** | **100.0**/ 92.6 / 99.2 / 99.9 /**100.0** |
| CIFAR-100 | TIM | 46.3 / 63.2 / 81.5 / 97.1 / **98.1** | 67.0 / 61.5 / 77.2 / 94.1 / **94.4** |
| | LSUN | 47.1 / 65.2 / 85.2 / 98.0 / **98.8** | 73.4 / 62.4 / 79.3 / 94.5 / **94.7** |
| | iSUN | 46.9 / 63.8 / 83.0 / 97.5 / **98.7** | 70.0 / 61.7 / 78.2 / 94.7 / **95.1** |
| | Gaussian | **100.0**/ 60.2 / 89.0 / 99.4 /**100.0** | **100.0**/ 84.7 / 99.3 /**100.0/100.0** |

## 6 CONCLUSION AND FUTURE WORK

In this paper, we demonstrated that the uncertainties of features extracted in each hidden layer are important for detecting OOD samples. We combined these uncertainties to obtain state-of-the-art OOD detection performance on several models and datasets. The approach proposed in this paper has the potential to increase the safety of many classification systems by improving their ability to detect OOD samples. In future work, our model could be used in an unsupervised model by training it to minimize reconstruction error, which would avoid the need to use in-distribution labels to detect OOD samples. Furthermore, although we compared our model with ODIN, UFEL will perform better if we combine UFEL with ODIN because they are orthogonal methods.

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

## A    DETAILS OF EXPERIMENTAL SETTINGS

### A.1    EVALUATION METRIC

We used several standard metrics for testing the detection of OOD samples and the classification of in-distribution samples.

**TNR at 95% TPR**. Let TP, TN, FP, and FN denote the numbers of true positives, true negatives, false positives, and false negatives, respectively. We measure TNR = TN / (FP+TN), when TPR = TP / (TP+FN) is 95%.

**AUROC**. The Area Under the Receiver Operating Characteristic curve (Davis & Goadrich, 2006) is a threshold-independent metric. The ROC curve depicts the relationship between TPR and FPR.

**AUPR**. The Area under the Precision-Recall curve is another threshold independent metric. The PR curve depicts the relationship between precision and recall. The metrics AUPR-In and AUPR-Out denote the area under the precision-recall curve in which in-distribution and OOD images are specified as positives, respectively.

**ACC**. We choose accuracy for measuring the classification ability of models when using in-distribution samples. We measure ACC = (TP+TN) / (TP + FP + TN + FN).

### A.2    DATASET

We used several standard datasets for detecting OOD samples and classifying in-distribution samples.

**CIFAR.** The CIFAR dataset (Krizhevsky et al., 2009) contains $32 \times 32$ natural color images. The training set has 50,000 images and the test set has 10,000 images. CIFAR-10 has 10 classes, whilst CIFAR-100 has 100 classes.

**SVHN.** The Street View Housing Numbers (SVHN) dataset (Netzer et al., 2011) contains $32 \times 32$ color images of house numbers. The training set has 604,388 images and the test set has 26,032 images. SVHN has 10 classes comprising the digits 0–9.

**TIM.** The Tiny ImageNet dataset consists of a subset of ImageNet images (Deng et al., 2009). It contains 10,000 test images from 200 different classes. We downsampled the images to $32 \times 32$ pixels.

**LSUN.** The Large-scale Scene UNderstanding (LSUN) dataset (Yu et al., 2015) has 10,000 test images of 10 different scenes. We downsampled the images to $32 \times 32$ pixels.

**iSUN.** The iSUN (Xu et al., 2015) dataset consists of a subset of 8,925 SUN images. We downsampled the images to $32 \times 32$ pixels.

**Gaussian Noise.** The Gaussian noise dataset consists of 10,000 random two-dimensional Gaussian noise images, where each value of every pixel is sampled from an i.i.d Gaussian distribution with mean 0.5 and unit variance.

**Uniform Noise.** The uniform noise dataset consists of 10,000 images, where each value of every pixel is independently sampled from a uniform distribution on [0, 1].

## A.3  MODEL STRUCTURE

| Layers | In channels | Out channels | Ksize | Stride | Padding |
|--------|-------------|--------------|-------|--------|---------|
| Conv2d | 3 | 64 | 5 | 1 | 0 |
| ReLU | - | - | - | - | - |
| MaxPool2d | - | - | - | 2 | - |
| Conv2d | 64 | 128 | 5 | 1 | 0 |
| ReLU | - | - | - | - | - |
| MaxPool2d | - | - | - | 2 | - |
| Linear | 128*5*5 | 120 | - | - | - |
| ReLU | - | - | - | - | - |
| Linear | 120 | 84 | - | - | - |
| ReLU | - | - | - | - | - |
| Linear | 84 | 10 | - | - | - |
| softmax | - | - | - | - | - |

Table 4: Network architecture of LeNet5.

| Layers | In channels | Out channels | Ksize | Stride | Padding |
|--------|-------------|--------------|-------|--------|---------|
| Conv2d | 128 | 64 | 4 | 1 | 2 |
| BatchNorm2d | - | - | - | - | - |
| ReLU | - | - | - | - | - |
| MaxPool2d | - | - | - | 2 | - |
| Conv2d | 64 | 1 | 4 | 1 | 2 |
| BatchNorm2d | - | - | - | - | - |
| ReLU | - | - | - | - | - |
| MaxPool2d | - | - | - | 2 | - |
| Feature Combination | 1*3*3 | 1*3*3 + y_dim + 1 | - | - | - |
| Linear | 1*3*3 + y_dim + 1 | 100 | - | - | - |
| BatchNorm1d | - | - | - | - | - |
| ReLU | - | - | - | - | - |
| Linear | 100 | 1 | - | - | - |
| Sigmoid | - | - | - | - | - |

Table 5: Network architecture of the feature-combining CNN for LeNet5.

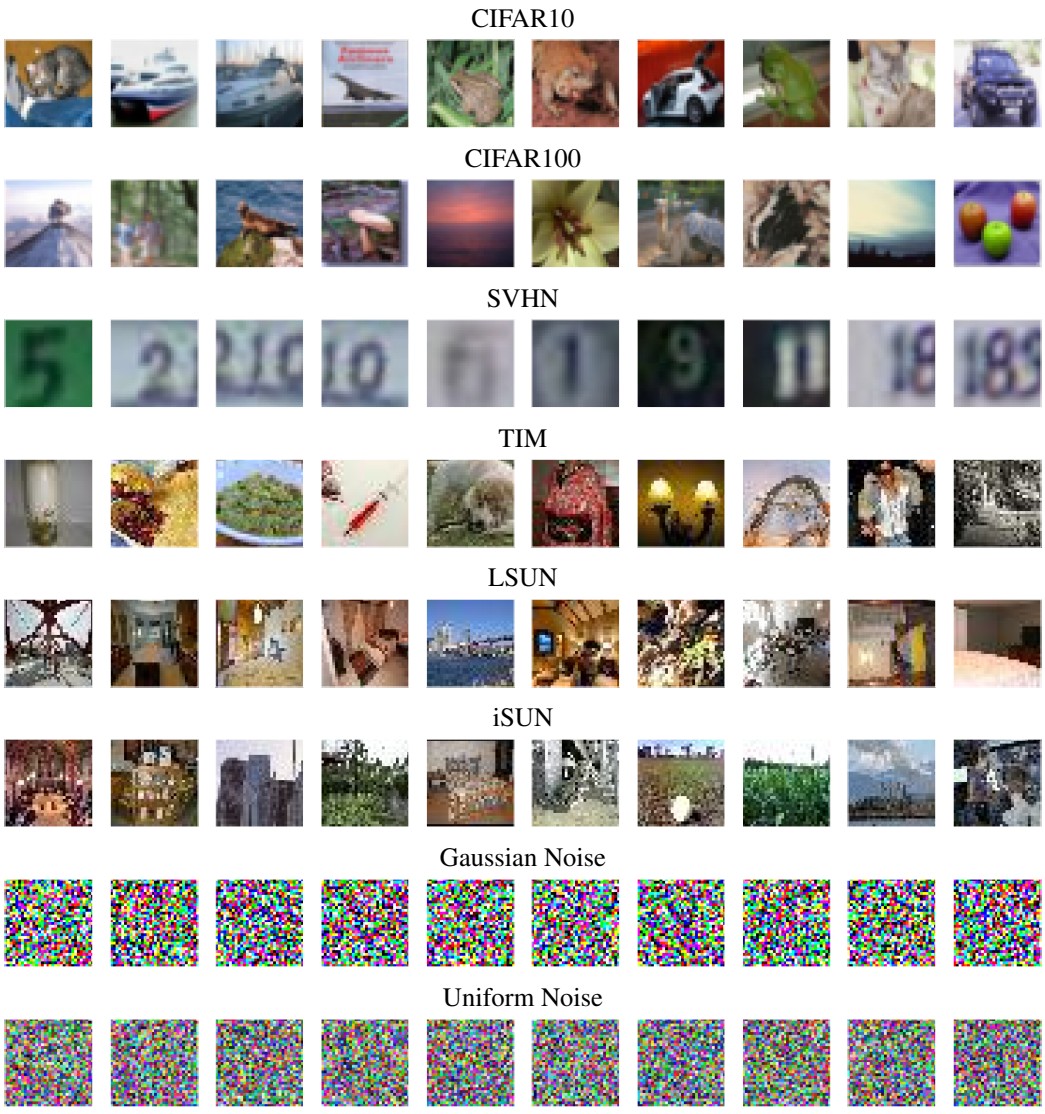

Figure 6: Visualization of original images.

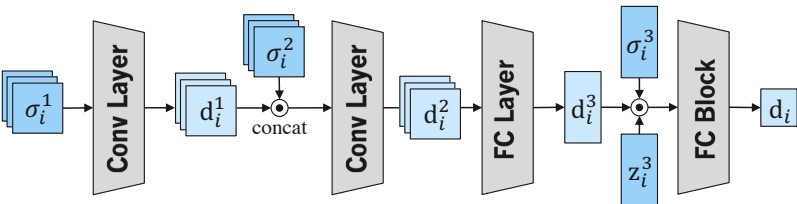

Figure 7: CNN for feature combination.

| Layers | In channels | Out channels | Ksize | Stride | Padding |
|---|---|---|---|---|---|
| Conv2d | 108 | 150 | 4 | 1 | 2 |
| BatchNorm2d | - | - | - | - | - |
| ReLU | - | - | - | - | - |
| MaxPool2d | - | - | - | 2 | - |
| Feature Combination | 150 | 150*2 | - | - | - |
| Conv2d | 150*2 | 64 | 4 | 1 | 2 |
| BatchNorm2d | - | - | - | - | - |
| ReLU | - | - | - | - | - |
| MaxPool2d | - | - | - | 2 | - |
| Conv2d | 64 | 1 | 4 | 1 | 2 |
| BatchNorm2d | - | - | - | - | - |
| ReLU | - | - | - | - | - |
| MaxPool2d | - | - | - | 2 | - |
| Feature Combination | 1*2*2 | 1*2*2 + y_dim + 1 | - | - | - |
| Linear | 1*2*2 + y_dim + 1 | 1024 | - | - | - |
| BatchNorm1d | - | - | - | - | - |
| ReLU | - | - | - | - | - |
| Linear | 1024 | 1 | - | - | - |
| Sigmoid | - | - | - | - | - |

Table 6: Network architecture of the feature-combining CNN for Dense-BC.

| Layers | In channels | Out channels | Ksize | Stride | Padding |
|---|---|---|---|---|---|
| Conv2d | 64 | 128 | 4 | 1 | 2 |
| BatchNorm2d | - | - | - | - | - |
| ReLU | - | - | - | - | - |
| MaxPool2d | - | - | - | 2 | - |
| Feature Combination | 128 | 128*2 | - | - | - |
| Conv2d | 128*2 | 64 | 4 | 1 | 2 |
| BatchNorm2d | - | - | - | - | - |
| ReLU | - | - | - | - | - |
| MaxPool2d | - | - | - | 2 | - |
| Conv2d | 64 | 1 | 4 | 1 | 2 |
| BatchNorm2d | - | - | - | - | - |
| ReLU | - | - | - | - | - |
| MaxPool2d | - | - | - | 2 | - |
| Feature Combination | 1*4*4 | 1*4*4 + y_dim + 1 | - | - | - |
| Linear | 1*4*4 + y_dim + 1 | 1024 | - | - | - |
| BatchNorm1d | - | - | - | - | - |
| ReLU | - | - | - | - | - |
| Linear | 1024 | 1 | - | - | - |
| Sigmoid | - | - | - | - | - |

Table 7: Network architecture of the feature-combining CNN for WRN-40-4.

