# OpenReview forum: "Out-of-Distribution Detection Using Layerwise Uncertainty in Deep Neural Networks"
_ICLR.cc/2020/Conference — Reject_

### Official Review · AnonReviewer1 · 2019-10-22
**Official Blind Review #1**

**Rating:** 1

**Review:**

This paper tackles out-of-distribution samples detection via training VAE-like networks. The key idea is to inject learnable Gaussian noise to each layer across the network in the hope that the variance of the noise correlates well with the uncertainty of the input features. The network is trained to minimize the empirical loss subject to noise perturbation. The paper is well written, and the background is introduced clearly.

As I understand it, the goal of *out-of-distribution sample detection* is to train a deep network that simultaneously generalizes well and also be discriminative to outliers. However, it’s not clear to me why the proposed method server this purpose; empirical results are not convincing either. My major concerns are as follows:

First of all, from my intuition, it would be much easier to train deterministic networks than their counterparts with randomness. Empirically, researchers also often observe near-zero training loss for large deterministic networks such as Dense-BC trained on simple CIFAR/SVHN datasets. Especially, in this case, the training goal is simply to map higher-dimensional inputs to lower-dimensional classification categories. That being said, one would expect the variances go to zero at convergence to achieve lower empirical loss in the case of no additional diversity (or uncertainty) promotion terms.

It is not clear to me how to avoid degenerate solutions at convergence
while maintaining good testing performance with the proposed training strategy.
From the empirical results, it also appears that all models reported might not be fully optimized?
The baseline results are significantly worse than those reported in previous work.
Specifically,
in table 1, the testing accuracy of Dense-BC trained on CIFAR-100 is only 71.6.
In table 2, the reported testing accuracy on CIFAR-10 using Dense-BC is 92.4.

However, the results of DenseNet-BC (k=12, L=100, table 2) reported in the original paper are:
CIFAR10  94.0  (also leave 5K examples as validation set)
CIFAR100 75.9

Meanwhile, the reported accuracy of WRN-40-4 trained on CIFAR-10 and CIFAR-100 are 89.6 and 66.0, respectively. However, the corresponding baseline numbers in the original WRN paper are much higher,
CIFAR-10  95.03
CIFAR-100 77.11

Could the authors comment on that?

References:
Gao Huang, Zhuang Liu, Laurens van der Maaten, Kilian Q. Weinberger.
Densely Connected Convolutional Networks
https://arxiv.org/abs/1608.06993

Sergey Zagoruyko, Nikos Komodakis.
Wide Residual Networks.
https://arxiv.org/pdf/1605.07146.pdf


**Experience Assessment:**

I have read many papers in this area.

**Review Assessment: Checking Correctness Of Derivations And Theory:**

I carefully checked the derivations and theory.

**Review Assessment: Checking Correctness Of Experiments:**

I carefully checked the experiments.

**Review Assessment: Thoroughness In Paper Reading:**

I read the paper thoroughly.

---

> ### Author Response · Authors · 2019-11-12
> **Response to Reviewer #1**
>
> Thank you for reviewing my paper.
>
> > That being said, one would expect the variances go to zero at convergence to achieve lower empirical loss in the case of no additional diversity (or uncertainty) promotion terms.
>
> This is true for the training data, but the variance term does not reduce to zero for the test data.
> In the discriminative space close to the output layer, the variance of OOD is larger than that of ID, thus we can detect OOD using this phenomenon.
>
> > It is not clear to me how to avoid degenerate solutions at convergence while maintaining good testing performance with the proposed training strategy.
>
> According to the Eq. (17) of [Alemi, 2016], the performance of classification is better, when the \beta is non-zero, namely there exists a regularization term. However, if we set \beta to a value that is close to zero, the model approaches the original deterministic function. Therefore, the performance will not be worse.
>
> > From the empirical results, it also appears that all models reported might not be fully optimized? The baseline results are significantly worse than those reported in previous work.
>
> We cannot access the test data to validate our method, thus we used 5,000 validation images taken from the 50,000 training data. This degenerates the performance of the classification. Therefore, the experimental conditions of the results reported in the paper you mention differ from those in our study. We confirmed that if we use all 50,000 training data, we obtain the same results as reported in the papers [He, 2016; Huang, 2017].
>
> [reference]
> Alexander A Alemi, Ian Fischer, Joshua V Dillon, and Kevin Murphy. Deep variational information
> bottleneck. arXiv preprint arXiv:1612.00410, 2016.

---

### Official Review · AnonReviewer3 · 2019-10-23
**Official Blind Review #3**

**Rating:** 3

**Review:**

** post rebuttal start **

After reading reviews and authors' response, I decided not to change my score.
However, I feel that this paper is somewhat under-evaluated initially, so I hope the authors have an opportunity in another venue with their revision.


Detailed comments:

1.1. I recommend to add an algorithm box describing the learning scheme. It is not end-to-end learning, so it is hard to catch (and potentially, replicate) the learning part. I am also a bit skeptical about the convergence (with non-zero \sigma), as Reviewer 2 has a concern about it.

1.3. "We hypothesized that the value of the uncertainty is different depending on whether the inputs are OOD or in-distribution inputs. The results of the ablation study listed in Table1 demonstrate that this hypothesis is true."
2. "In order to use the data uncertainty, we used the value of \sigma."
-> Table 1 proves that your proposal (playing with \sigma) is effective, but it does not mean that \sigma is the uncertainty which is only essential component for detecting OOD. I recommend the authors to validate their hypothesis, maybe by conducting more experiments to show that the role of \mu and \sigma is as expected. At least, if \mu is proven to have no effect on OOD detection by some experiment, then it can be a clue.


Minor comment: I hope ICLR papers are cited as ICLR papers at least in ICLR submissions, not arXiv preprint.. Alemi's paper is ICLR'17 paper, for example.

** post rebuttal end **



- Summary: This paper proposes to train an OOD detection model from a portion of modified latent vectors; more specifically, similar to VAE, they assume unimodal Gaussian distributed latent space at each layer and use the collection of standard deviation to train an OOD detector. Experimental results on several OOD benchmarks with different backbone networks show that their method outperforms ODIN (Liang et al., 2017).


- Decision and supporting arguments:
Weak reject.

1. Though the idea of extracting uncertainty is interesting, but I think the motivation and explanation is not enough, so I couldn't find a rationale why we should do this. I have several questions that I couldn't find an answer in the submission, could you answer them?
1.1. Are the classification loss and OOD detection loss optimized jointly?
1.2. Is it reasonable to assume unimodal Gaussian distribution over all latent spaces without a carefully designed learning objective? More specifically, to make it learnable, don't you need a learning objective other than the conventional cross-entropy loss, e.g., "Bayes by backprop" proposed in the early work (Blundell, 2015)?
1.3. Why only the standard deviation values are useful for the OOD detection performance? If they are really useful, how the standard deviation values are related to the OOD detection performance?

Blundell et al. Weight Uncertainty in Neural Networks. In ICML, 2015.

2. More ablation study is required to verify the effectiveness of their method. Again, I am not sure why \mu and \sigma should be split, and why \mu should be discarded for the OOD detection part.

3. The architecture design of CNN in Figure 7 also looks arbitrary.

4. Comparison with more state-of-the-art methods is required. ODIN (Liang et al., 2017) is a powerful method but it is somewhat old and many recent works actually combine their method with ODIN for better performance. Why don't you compare the proposed method with the Mahalanobis distance-based classifier (Lee et al., 2018)? They also estimate the uncertainty by measuring the Mahalanobis distance on the feature spaces & combine them for better OOD detection.


- Comments:
1. I couldn't find any statement about the classification accuracy, does the proposed model have a good classification performance as well? Since {a half of the model capacity is spent to split \mu and \sigma} and {it should take account of uncertainty in the forward pass}, I am not sure it maintains a good classification performance, compared to the standard classification model with the same capacity.

**Experience Assessment:**

I have published in this field for several years.

**Review Assessment: Checking Correctness Of Derivations And Theory:**

I assessed the sensibility of the derivations and theory.

**Review Assessment: Checking Correctness Of Experiments:**

I assessed the sensibility of the experiments.

**Review Assessment: Thoroughness In Paper Reading:**

I read the paper at least twice and used my best judgement in assessing the paper.

---

> ### Author Response · Authors · 2019-11-12
> **Response to Reviewer #3**
>
> Thank you for your review.
>
> > 1.1. Are the classification loss and OOD detection loss optimized jointly?
>
> No, we optimized those losses separately.
>
> > 1.2. Is it reasonable to assume unimodal Gaussian distribution over all latent spaces without a carefully designed learning objective? More specifically, to make it learnable, don't you need a learning objective other than the conventional cross-entropy loss, e.g., "Bayes by backprop" proposed in the early work (Blundell, 2015)?
>
> There are some Bayesian methods that obtain the uncertainty, but the uncertainty is basically a type of model uncertainty. We cannot detect OOD samples using model uncertainty because the uncertainty will be larger, even if the inputs are in-distribution samples, because of the lack of training data. According to [Malinin & Gales, 2018], the performance of Bayesian methods is not good. Therefore, we used reparameterization to employ the data uncertainty, referring to the approach in [Alemi, 2016].
>
> > 1.3. Why only the standard deviation values are useful for the OOD detection performance? If they are really useful, how the standard deviation values are related to the OOD detection performance?
>
> We do this because we would like to use the data uncertainty of the intermediate layer and the last layer by using reparameterization, as in VIB [Alemi, 2016]. We hypothesized that the value of the uncertainty is different depending on whether the inputs are OOD or in-distribution inputs. The results of the ablation study listed in Table1 demonstrate that this hypothesis is true. This result shows that the sum of each variance is a feature that can discriminate between OOD and in-distribution inputs. This is because the variance of the feature close to the input layer is larger when the inputs are in-distribution samples, whereas the variance of the output close to the output layer is larger when the inputs are OOD samples.
>
> > 2. More ablation study is required to verify the effectiveness of their method. Again, I am not sure why \mu and \sigma should be split, and why \mu should be discarded for the OOD detection part.
>
> In order to use the data uncertainty, we used the value of \sigma.
>
> > 3. The architecture design of CNN in Figure 7 also looks arbitrary.
>
> The architecture shown in Figure 7 is an example of a CNN. Any model whose inputs consist of feature maps can be used.
>
> > 4. Comparison with more state-of-the-art methods is required. ODIN (Liang et al., 2017) is a powerful method but it is somewhat old and many recent works actually combine their method with ODIN for better performance. Why don't you compare the proposed method with the Mahalanobis distance-based classifier (Lee et al., 2018)? They also estimate the uncertainty by measuring the Mahalanobis distance on the feature spaces & combine them for better OOD detection.
>
> We will experiment with a Mahalanobis distance-based classifier in addition to our current results.
>
> > Comments: 1. I couldn't find any statement about the classification accuracy, does the proposed model have a good classification performance as well? Since {a half of the model capacity is spent to split \mu and \sigma} and {it should take account of uncertainty in the forward pass}, I am not sure it maintains a good classification performance, compared to the standard classification model with the same capacity.
>
> According to the Eq. (17) of [Alemi, 2016], the performance of classification is better, when the \beta is non-zero, namely there exists a regularization term. However, if we set \beta to a value that is close to zero, the model approaches the original deterministic function. Therefore, the performance will not be worse. As the results in Table 2 in our paper confirm, the classification of the performance is not worse.
>
> [reference]
> Alexander A Alemi, Ian Fischer, Joshua V Dillon, and Kevin Murphy. Deep variational information
> bottleneck. arXiv preprint arXiv:1612.00410, 2016.
>
> Andrey Malinin and Mark Gales. Predictive uncertainty estimation via prior networks. In Advances in Neural Information Processing Systems, pp. 7047–7058, 2018.

---

### Official Review · AnonReviewer2 · 2019-10-23
**Official Blind Review #2**

**Rating:** 1

**Review:**

The paper considers the problem of out-of-distribution (OOD) sample detection while solving a classification task. The authors tackle the problem of OOD detection with exploiting uncertainty while passing a test sample through the neural network. They treat outputs of (some) layers in a NN as random Gaussian-distributed variables and measure uncertainty as variance of these Gaussians. Then when uncertainty is high, OOD is detected.

The overall idea behind the paper could be interesting, but its realisation in the current form is questionable.

The paper seems totally misusing the reparametrisation trick and stochastic outputs of layers in NNs. Eq. (2) is not the objective of variational inference that seems to be required for stochastic outputs and the reparametrisation trick as presented before the equation. The objective misses the KL-divergence term! Without it what would stop a neural net to set sigmas to 0 and forget about the stochasticity altogether? Not to mention that the current objective is not mathematically justified.
If there is no mix and error in eq. (2) and the networks were trained using this loss (and based on provided code they were using this loss), my wild guess of explaining why this may give best results in the experiments is that the models were trained for surprisingly small number of epochs. Therefore, a hypothesis would be that this small number of epochs did not allow the networks to switch sigmas to 0.

Until the authors can clarify and justify the objective, I will vote for rejection only based on this ground.

However, there are other issues in the paper as well. First of all, its clarity. It seems that the paper requires a lot of polishing.  The first paragraph of this review is based on my assumptions from the paper since I am not completely sure I understand it correctly. More about the clarity issues below

For strong evaluation, comparison with Malinin & Gales (2018) work seems to be important since it was the only work also using uncertainties for OOD detection in related work. Also related work section does not look like an exhaustive overview.

Some of the detailed comments:
1.	“In other words, in-distribution samples possess more features that convolutional filters react to than OOD samples” – first of all, this sentence is not easy to parse. Secondly, it is unclear, why this should be true. If OOD samples are still natural images, they would contain edges just the same as in-distribution samples. That is the power of deep learning enabling transfer learning, that low-level features are the transferable across different data and tasks. Therefore, the claim that “Therefore, the uncertainties of the features will be larger when the inputs are in-distribution samples” requires more elaboration and arguments
2.	The arguments of the next paragraph regarding uncertainty of deeper layers should be larger for OOD samples are not very convincing either.  It is either requires a definition what the authors mean here as uncertainty, or it is not necessarily true that absence of fixed regions for embeddings leads to higher uncertainty.
3.	3rd and 4th paragraphs in Introduction have too many repetitions of phrases between each other. Compare, e.g. the first sentences of the paragraphs or the last sentences.
4.	“One cause of the abovementioned problem is that their approaches” and similarly the next paragraph: “their approaches” stylistically sound wrong. It is appropriate in the previous paragraph since there is a link to “previous studies”. It seems that “these approaches” or “the existing approaches” would be a better choice for this and the next paragraph.
5.	“Each uncertainty is easily estimated after training the discriminative model by computing the mean and the variance of their features using a reparameterization trick” – conventional discriminative models do not estimate mean and variances of the features. The issue of estimating uncertainty is addressed by several special methods such as Bayesian variational methods used in the referred papers. Therefore, in order to use a reparametrisation trick one need to firstly choose a special class of models, which is not obvious from the text.
6.	“Moreover, UFEL is robust to hyperparameters such as the number of in-distribution classes and the validation dataset.” – the size of the validation dataset? In any case neither size of the validation dataset nor the validation dataset itself are not hyperparameters (should not be hyperparameters for out-of-distribution detection). The number of classes can hardly be called a hyperparameter also.
7.	“depends on the difference in the Dirichlet distribution of the categorical parameter <…> In our work, the distribution of the logit of the categorical parameters” – what is/are this/these categorical parameter(s)?
8.	“Further, they estimate the parameter of the Dirichlet distribution using a DNN and train the model with in-distribution and OOD datasets” – this sentence may mislead to impression that the proposed method does not need OOD dataset for training, which does not seem to be the case, since \lambda and \theta are trained based on OOD samples
9.	“because they will not be relevant to the classification accuracy” – who are they?
10.	“and \epsilon is the Gaussian noise” –> the standard Gaussian noise
11.	“where z^0 = x” – it seems this should be placed somewhere earlier when z^l is introduced since z^0 is not used in eq.(2) after which this text is placed
12.	It is unclear how \lambda^l and CNN \theta are learnt
13.	It is unclear how the values of features d(x) are used to detect OOD samples
14.	“comparison methods, and models” – not clear what models mean here
15.	Missing references to datasets in the main text. At least reference to Appendix A.2 is required
16.	“We used 5,000 validation images split from each training dataset and chose the parameter that can obtain” – which parameter?
17.	“All the hyperparameters of ODIN” – a reader does not know yet that ODIN is used for comparison
18.	“which consists of 100 OOD images from the test dataset and 1,000 images from the in-distribution validation set” – it is a bit confusing to call OOD dataset as a test dataset in this context
19.	“We tuned the parameters of the CNN in Equation 4 using 50 validation training images taken from the 100 validation images. The best parameters were chosen by validating the performance using the rest of 50 validation images.” – this is confusing. What parameters do the authors talk about in the second sentence if not the parameters of the CNN?
20.	“We used TNR at 95% TPR, AUROC, AUPR, and accuracy (ACC),” – Some elaboration is required, at least the reference to Appendix A.1. What is the changing threshold for AUROC and AUPR? Why AUPR-In and AUPR-Out are considered and only a single AUROC is considered. What is the positive class for AUROC?
23.	“For LeNet5, we increased the number of channels of the original LeNet5 to improve accuracy” – do the authors mean that they allowed RGB images as input rather than greyscale? If yes, this explicit explanation would be preferable
24.	“We inserted the reparameterization trick” – not the best word choice. Reparametrisation trick is a computational/implementation trick/method and it is hard to say that it can be inserted into a network. I believe what the authors mean is that they inserted mean/std outputs instead of point outputs. Conceptually, this means that the output of the corresponding layers is considered to be stochastic rather than deterministic.
Also, it is unclear when the authors say they insert it to the softmax layer. According to Section 3 the softmax layer is never considered to output means and stds.
25.	The numbers of epochs for training NNs are very small for LeNet and WideResNet in the experiments. Did the models manage to converge during this short training?

Minor:
1.	“These data were also used” -> “this data”

**Experience Assessment:**

I do not know much about this area.

**Review Assessment: Checking Correctness Of Derivations And Theory:**

I carefully checked the derivations and theory.

**Review Assessment: Checking Correctness Of Experiments:**

I assessed the sensibility of the experiments.

**Review Assessment: Thoroughness In Paper Reading:**

I read the paper thoroughly.

---

> ### Author Response · Authors · 2019-11-12
> **Response to Reviewer #2**
>
> Thank you so much for your very detailed comments!
>
> > The paper seems totally misusing the reparametrisation trick and stochastic outputs of layers in NNs. Eq. (2) is not the objective of variational inference that seems to be required for stochastic outputs and the reparametrisation trick as presented before the equation.The objective misses the KL-divergence term! Without it what would stop a neural net to set sigmas to 0 and forget about the stochasticity altogether?
>
> The Eq. (2) is not the objective of variational inference. We referenced the Eq. (17) of [Alemi, 2016], and we set \beta to zero in the method in our paper. As you state, the variance of the output approaches zero when we use training data. However, the variance of the output is not zero when using OOD data, and we assume that this helps us to detect OOD samples.
>
> > However, there are other issues in the paper as well. First of all, its clarity. It seems that the paper requires a lot of polishing.
>
> Thank you so much for your many comments. We plan to modify the main text soon.
>
> > 1. “In other words, in-distribution samples possess more features that convolutional filters react to than OOD samples” – first of all, this sentence is not easy to parse. Secondly, it is unclear, why this should be true. If OOD samples are still natural images, they would contain edges just the same as in-distribution samples. That is the power of deep learning enabling transfer learning, that low-level features are the transferable across different data and tasks. Therefore, the claim that “Therefore, the uncertainties of the features will be larger when the inputs are in-distribution samples” requires more elaboration and arguments
>
> As you state, the outputs close to the input layer will be similar whether the input is an in-distribution or OOD sample. However, the output becomes more specific to in-distribution samples as the layers become deeper. Furthermore, when we use transfer learning, we may not use a layer that is close to the output layer. However, the outputs of the layers that are too close to the output layer are sometimes incorrectly regarded as OOD, as we mentioned in our paper. Therefore, we use the feature of an intermediate layer to improve the performance of OOD detection.
>
> > 5. “Each uncertainty is easily estimated after training the discriminative model by computing the mean and the variance of their features using a reparameterization trick” – conventional discriminative models do not estimate mean and variances of the features. The issue of estimating uncertainty is addressed by several special methods such as Bayesian variational methods used in the referred papers. Therefore, in order to use a reparametrisation trick one need to firstly choose a special class of models, which is not obvious from the text.
>
> There are some Bayesian methods that obtain the uncertainty, but the uncertainty is basically a type of model uncertainty. We cannot detect OOD samples using model uncertainty because the uncertainty will be larger, even if the inputs are in-distribution samples, because of the lack of training data. According to [Malinin & Gales, 2018], the performance of Bayesian methods is not good. Therefore, we used reparameterization to employ the data uncertainty, referring to the approach in [Alemi, 2016].
>
> [reference]
> Alexander A Alemi, Ian Fischer, Joshua V Dillon, and Kevin Murphy. Deep variational information
> bottleneck. arXiv preprint arXiv:1612.00410, 2016.
>
> Andrey Malinin and Mark Gales. Predictive uncertainty estimation via prior networks. In Advances in Neural Information Processing Systems, pp. 7047–7058, 2018.

---

> > ### Comment · AnonReviewer2 · 2019-11-12
> > **Eq. (2) is still unclear**
> >
> > Thank you for your reply, but not all your answers quite clarify my concerns.
> >
> > My main concern with the paper was and still is the eq. (2). The quote from Alemi at el: "(Note that setting beta = 0 corresponds to the usual maximum likelihood estimate.)". Therefore, you agree you are using the maximum likelihood estimate. Then, the choice of treating the outputs of the layers as random variables and yet using maximum likelihood estimates for weights of the neural network is at least unconventional. And it is definitely requires more explanation, motivation and justification.
> >
> > Again, with the current objective, what would stop the neural network to learn sigmas as zeros and to not deal with stochasticity at all. I meant sigmas constant zero functions, e.g. all weights of f^{\Sigma}_{\phi_l} = 0. Therefore, your comment on variance for OOD data being not zero does not answer my concern, sorry for misunderstanding.

---

> > > ### Author Response · Authors · 2019-11-15
> > > **Response to the comments**
> > >
> > > Thank you for your reply.
> > >
> > > >  I meant sigmas constant zero functions, e.g. all weights of f^{\Sigma}_{\phi_l} = 0.
> > >
> > > As you state, the variance of output approaches zero, but it does not always become zero. For example, when the mean of output moves far from a decision boundary, the variance does not need to become zero. In the experiment, we confirmed that the value of the variance is very small, but not zero. In addition, the classification accuracy of our model is slightly higher than that of the regular model. We can therefore consider that the mean of the output might move far from the decision boundary.

---

> > > > ### Comment · AnonReviewer2 · 2019-11-15
> > > > **Response to the response**
> > > >
> > > > Thank you for your response.
> > > >
> > > > I am interested in a principled way - what would prevent a neural network to learn all sigmas as zeros.
> > > > Unfortunately, the experiments in the paper are not too convincing that "it does not always become zero" due to the very limited number of epochs used in the experiments, as I raised in the initial review. As you are saying if the variance approaches zero and if we increase the number of epochs (to the number more consistent with the other works), would the variance become exact zero?
> > > >
> > > > That is the important question but this is just a part of a bigger question I concern about the model. The objective without the KL-term is unconventional. This is not a bad thing, it is a very good thing if it is actually better and it would open a new direction of research. However, the objective with the KL-term is mathematically grounded. Therefore, the objective without it requires much more introduction, intuition and justification than it is currently presented in the paper. If there is no yet theoretical justification for this objective, but "it just works" in practice, it may be also ok, but it would require more convincing experiments (for example, to resolve any doubts that the only reason sigmas do not approch zeros is because of the very limited number of epochs).

---

### Decision · Program_Chairs · 2019-12-19

**Decision:**

Reject

**Comment:**

The paper proposes a method for OOD detection which leverages the uncertainties associated with the features at the intermediate layers (and not just the output layer).

All the reviewers agreed that while this is an interesting direction, the paper requires more work before it can be accepted. In particular, the reviewers raised several concerns about other relevant baselines, some of the reported empirical results, and clarity of the explanation.

I encourage the authors to revise the draft based on the reviewers’ feedback and resubmit to a different venue.